# Potential risk factors associated with seropositivity for *Toxoplasma gondii* among pregnant women and HIV infected individuals in Ethiopia: A systematic review and meta-analysis

Zewdu Seyoum Tarekegn[1]*, Haileyesus Dejene[2], Agerie Addisu[3], Shimelis Dagnachew[1]

**1** Department of Paraclinical Studies, College of Veterinary Medicine and Animal Sciences, University of Gondar, Gondar, Ethiopia, **2** Department of Veterinary Epidemiology and Public Health, College of Veterinary Medicine and Animal Sciences, University of Gondar, Gondar, Ethiopia, **3** Department of Biology, College of Natural and Computational Sciences, University of Gondar, Gondar, Ethiopia

* zewdusagera@gmail.com, zewdu.seyoum@uog.edu.et

## Abstract

### Background

*Toxoplasma gondii* is an obligate intracellular and neurotropic apicomplexan parasite infecting almost all warm-blooded vertebrates including humans. To date in Ethiopia, no systematic study has been investigated on the overall effects of potential risk factors associated with seropositivity for *Toxoplasma gondii* among pregnant women and HIV infected individuals. We intended to determine the potential risk factors (PRFs) associated with seropositivity for *Toxoplasma gondii* from published data among pregnant women and HIV infected individuals of Ethiopia.

### Methodology

An systematic review of the previous reports was made. We searched PubMed, Science Direct, African Journals Online, and Google Scholar for studies with no restriction on the year of publication. All references were screened independently in duplicate and were included if they presented data on at least two risk factors. Meta-analysis using the random or fixed-effects model was made to calculate the overall effects for each exposure.

### Results

Of the 216 records identified, twenty-four reports met our eligibility criteria, with a total of 6003 individuals (4356 pregnant women and 1647 HIV infected individuals). The pooled prevalences of anti-*Toxoplasma gondii* antibodies were found at 72.5% (95% CI: 58.7% - 83.1%) in pregnant women and 85.7% (95% CI: 76.3% - 91.8%) in HIV infected individuals. A significant overall effect of anti-*Toxoplasma gondii* seropositivity among pregnant women ($p < 0.05$) was witnessed with age, abortion history, contact with cats, cat ownership, having knowledge about toxoplasmosis, being a housewife and having unsafe water source. Age,

**Data Availability Statement:** All relevant data are within the paper and its Supporting Information files.

**Funding:** The author(s) received no specific funding for this work.

**Competing interests:** The authors have declared that no competing interests exist.

cat ownership, and raw meat consumption were also shown a significant effect ($p < 0.05$) to anti-*Toxoplasma gondii* seropositivity among HIV infected individuals.

## Conclusions

This review showed gaps and drawbacks in the earlier studies that are useful to keep in mind to design accurate investigations in the future. The pooled prevalence of anti-*Toxoplasma gondii* antibodies was found to be higher among pregnant women and HIV infected individuals. This suggests that thousands of immunocompromised individuals (pregnant women and HIV infected patients) are at risk of toxoplasmosis due to the sociocultural and living standards of the communities of Ethiopia. Appropriate preventive measures are needed to reduce the exposure to *Toxoplasma gondii* infection. Further studies to investigate important risk factors are recommended to support the development of more cost-effective preventive strategies.

## Author summary

*Toxoplasma gondii* is a cosmopolitan food and water-borne zoonotic protozoan parasite that able to infect almost all warm blooded vertebrates. It causes a considerable public health impact with higher burden in developing countries. Estimating pooled prevalence of *Toxoplasma* infection in high risk groups can draw health policy makers' attention in planning a screening program and control needs. Several risk factors determine the circulation of *Toxoplasma gondii* between intermediate and final hosts. In this comprehensive systematic review and meta-analysis, we intended to determine the prevalence and associated potential factors of *Toxoplasma gondii* infection in pregnant women and HIV infected individuals of Ethiopia. We searched English electronic databases (PubMed, Science Direct, African Journals Online, and Google Scholar) for studies. Our review resulted in a total of 24 papers meeting the inclusion criteria. The studies were performed from 2007 to 2019 using a cross-sectional study design and with overall samples of 6003 individuals (4356 pregnant women and 1647 HIV infected individuals). The estimated pooled seroprevalence of anti-*Toxoplasma gondii* antibodies using random-effect model was found to be 72.5% (95% CI: 58.7% - 83.1%) in pregnant women and 85.7% (95% CI: 76.3% - 91.8%) in HIV infected individuals. Risk factors: age, abortion history, contact with cats, cat ownership, having knowledge about toxoplasmosis, being a housewife and having unsafe water source were showed significant overall effect on *Toxoplasma gondii* infection rate in pregnant women. Age, cat ownership and raw meat consumption also showed significant overall effect on infection rate with *Toxoplasma gondii* in HIV infected individuals. The findings of this study are valuable to increase awareness among public health workers and educators regarding *Toxoplasma gondii* infection in high risk groups (pregnant women and HIV infected individuals) and risk factors that influence the occurrence of its infection. The findings also indicate a need for prenatal screening for early diagnosis and treatment.

## Introduction

*Toxoplasma gondii* is an obligate intracellular and neurotropic apicomplexan parasite infecting almost all warm-blooded vertebrates including humans. It causes serious and

life-threatening diseases in immunodeficient individuals and developing fetuses [1]. Globally, it has been estimated to infect approximately 30% of the human population and cause considerable public health impacts with a higher burden in developing countries [2].

Food animals with infected tissue cysts are important sources of human infection. Humans can acquire the infection through three main pathways: 1) by eating raw or undercooked meat harbouring viable tissue cysts; 2) by ingesting contaminated water and food or soil with oocysts shed by cats to the environment and 3) congenital transmission from infected mother to the foetus during pregnancy [3–6]. Domestic and wild felids play an important role in the epidemiology of *Toxoplasma gondii* because they are the only definitive hosts capable of excreting viable oocysts in their faeces [3]. They become infected with *Toxoplasma gondii* by eating infected tissues from intermediate hosts.

Infection in healthy individuals is usually asymptomatic and often results in the chronic persistence of cysts within host tissues [5]. But in immunodeficient patients (like HIV/AIDS patients), it can lead to life-threatening encephalitis owing to reactivation of latent infections [4,5,7,8]. Further, infection of pregnant women may result in miscarriage or spontaneous abortion or congenital infection that may cause severe pathological defects (hydrocephalus, foetal death, deafness, blindness, mental retardation, or neurological damage, intracranial calcification and retinochoroiditis) [7,8]. It has also been reported as a possible risk factor for personality shifts, epilepsy, bipolar disorder, a suicide attempt, car accident, and reduced intelligence or schizophrenia [9,10].

The epidemiology of *Toxoplasma gondii* infection in immunocompromised individuals (pregnant women and HIV infected patients) showed considerable variation between continents and countries. Recent systematic reviews have assessed and estimated the global pooled prevalence of anti-*Toxoplasma gondii* antibodies ranges from 1.1% to 33.8% in pregnant women [11–13] and 3.24% to 44.22% in HIV infected individuals [1,14]. In Africa, the pooled infection rate ranges from 1.6% to 48.7% in pregnant women [13] and 0.61% to 44.9% in HIV infected individuals [1]. In Ethiopia, based on empirical data, the infection rate estimates of 18.5% - 96.3% have been reported in the different risk groups of the population [4,15,16].

Various risk factors (the socioeconomic status and cultural habits of the community, health care education and economic status, geographical factors, cat lifestyle and density, wildlife structure, climate conditions, the virulence and genotype of the parasite and mode of transmission) have been documented to affect the host-pathogen interaction [3,17,18]. Thus, control options for *Toxoplasma gondii* infection must rely on the vigorous evidence of the risk factors contributing to its circulation among hosts [19]. However, the relative effects of each noted factors have not been fully summarized, except their variations from area to area [17]. In Ethiopia, despite high public health significance of *Toxoplasma gondii* along with extensive practices of raw meat consumption, keeping of cat as pet animals, presence of stray cats and suitable climatic conditions maintaining the survival of the pathogen, no systematically organized reference data that can be used by policy makers and public health authorities to plan strategic prevention and control measures. Besides, due to the presence of fragmented reports, there is limited knowledge on the trend and prevalence with associated risk factors of *Toxoplasma gondii* infection in pregnant women and HIV infected individuals. A better understanding of the risk factors associated with the prevalence of *Toxoplasma gondii* infection is important to plan and establish prevention measures. Therefore, to address these limitations and to draw the attention of researchers, public health authorities, stakeholders, policy makers and governments towards the public health importance of *Toxoplasma gondii*, this systematic review and meta-analysis was conducted to determine the potential risk factors (PRFs) associated with seropositivity for *Toxoplasma gondii* from published data among pregnant women and HIV infected individuals of Ethiopia.

## Methods

### Search strategy

Literature searching was performed following the Preferred Reporting Items for Systematic Reviews and Meta-analyses (PRISMA) guidelines (S1 Text) [20]. We searched PubMed, Science Direct, African Journals Online, and Google scholar databases with no restriction on the year of publication up to 30th November 2019. Article search was made using the searching terms: (Toxoplasmosis OR *Toxoplasma* infection OR *Toxoplasma gondii* OR *T. gondii* OR *Toxoplasma*) AND (Seroprevalence OR Prevalence OR Seroepidemiology) AND (Risk factors OR Potential factors) AND (Pregnant women) AND (Ethiopia). Also, keywords: (HIV infected individuals OR HIV/AIDS patients) were used. Moreover, unpublished thesis manuscripts were also accessed from Ethiopian Universities.

### Inclusion criteria

We used the following inclusion criteria to confirm the eligibility of the searched papers: (1) original research articles and theses; (2) cross-sectional, case-control and cohort studies that were reported seroprevalence and risk factors; (3) studies with full texts; (4) targeted study population: pregnant women and HIV infected individuals of Ethiopia; (5) studies with serological tests; (6) studies that provided the total sample size and the outcome of interest; and (7) studies published in the English language. Each paper that did not meet the above-mentioned criteria was excluded.

### Data extraction

Initially, articles were screened based on their titles and abstracts following the predefined inclusion criteria. Then, articles that seem potential for eligibility were selected and downloaded in full text. The searched articles were reviewed and abstracted carefully by two independent reviewers to prove eligibility. Disagreements between reviewers were settled by discussion. For each article, the following information were extracted: first author, publication year, study year, location, study design, sampling method, sample size, study subject, diagnostic test, potential risk factors (exposure), number of positive and negative samples. Study effect size, the odds ratio, and their corresponding confidence intervals were also calculated from the extracted data (S1 Data).

Data on potential risk factors (PRF) such as place of residence, contact with a cat, ownership of a cat or dog, water source, consumption of raw meat, vegetables and milk, age (15–34 versus ≥35 years in pregnant women and ≥25 versus < 25 years in HIV infected individuals), occupational group, education level, soil contact, number of pregnancies and history of abortion, and gestation period were recorded. Study searching strategies and exclusion criteria are presented in detail in Fig 1. Mendeley version 1.19.5 was used to catalogue the initial literature search results and to manage citations. Microsoft Excel datasheet was used to code and manage all extracted information from all relevant studies.

### Study quality assessment

The qualities of included papers were evaluated by two independent researchers using a quality assessment checklist (standard strengthening the Reporting of Observational Studies in Epidemiology checklist (STROBE) [21]. This quality assessment checklist includes 22 items constituting various sections of the articles such as title, abstract, introduction, methods, results, and discussion. The checklist included items assessing objectives, different components of the methods (e.g., study design, sample size, study population, bias, statistical methods), results,

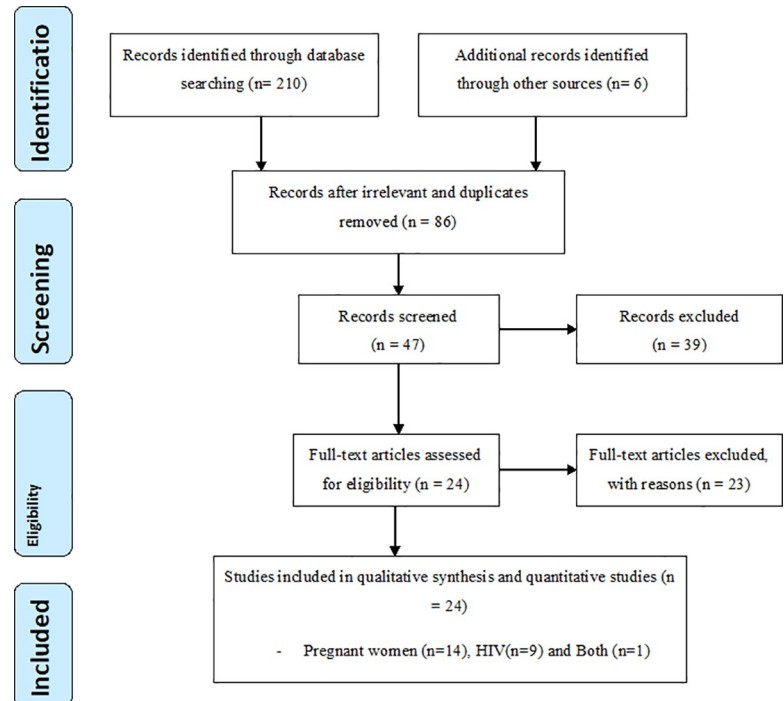

**Fig 1. PRISMA flow diagram used for study searching process.**

limitations, and funding of the studies. The assigned scores were determined from 0 to 44. Following the checklist (STROBE), searched papers were classified into 3 groups: low quality score (< 15.5), moderate quality (15.5–29.5), and high quality (30.0–44.0). S1 Table shows the checklist of the quality of the included studies [21].

## Meta-analysis

The meta-analysis was performed according to the protocol noted by DerSimonian and Laird [22]. In brief, data on pregnant women and HIV infected individuals were analyzed separately. The *Toxoplasma gondii* seroprevalence and its corresponding 95% confidence interval (CI) in pregnant women and HIV infected individuals were calculated for each study. Logit transformation was performed to transform study-level estimates to logit event estimates [23] using the formula: lp: ln [p/ (1-p)], where lp = logit event estimate; ln = natural logarithm; p = study level estimate/prevalence. The variance of the estimates was also computed using the formula: var (lp) = 1/np + 1/ (n-np), where var = variance and n = sample size. Then, the population/ weighed prevalence (WP) of *Toxoplasma gondii* infection in all study groups was calculated using the formula WP = $\Sigma$ (pi/var (pi)) /$\Sigma$ (1/var (pi)). The pooled prevalence estimate was computed using the formula, p = exp (lp)/ (exp (lp) +1), where exp (lp) = exponential of the logit event estimate [23]. Also, meta-analyses were made to determine the effects of each identified PRF if at least two included studies reported data on the same risk factor. OR and its respective 95% CI were calculated for each risk factor. We used pooled odds ratios (OR) as a measure of effect to assess the overall effects of each PRF. The forest plot was employed to present the outcomes of the meta-analysis. Cochran's Q-statistics and inverse variance index ($I^2$) were computed to determine the heterogeneity and inconsistency among studies, respectively [24]. We considered the $I^2$ values of 25, 50, and 75% as a low, medium, and high heterogeneity, respectively [25]. The *tau* statistics ($\tau^2$) were used to assess the variance of the effect size

estimates across the population of the study. Galbraith plot was also constructed to assess the heterogeneities of study level estimates. We used the random-effect model (if the *p*-value of the Q test was < 0.05 and $I^2$ was >50%) to pool the estimations following the heterogeneity result. Besides, we used a random-effect model for risk factor assessment if we had five and above studies to accept the previous cutoff points [26]. Further, the funnel plot, Egger's, and Begg's tests [27] were performed to assess small study effects and publication bias. STATA software version 16 (StataCorp, College Station, TX, USA) was used for meta-analyses.

## Results

### Search results and characteristics

We searched both published and unpublished (thesis) reports of *Toxoplasma gondii* infection in pregnant and HIV infected individuals in Ethiopia. Our literature search period was commenced from August 2019 to November 2019. We retrieved 216 reports for *Toxoplasma gondii* infection in pregnant and HIV infected individuals. Of them, 24 papers were found to be eligible for the inclusion criteria and data extraction (Fig 1). All selected studies were performed between 2007 and 2019 using a cross-sectional study design with convenient, systematic, and simple random sampling procedures. The studies were undertaken from various regions of Ethiopia: Addis Ababa, Amhara, Oromia, Southern Nations and Nationalities of Peoples, Somali and Tigray. Further, the studies were performed using a latex agglutination test (LAT), Enzyme-linked immunosorbent assay (ELISA), and Enzyme immune assay (EIA) to detect anti-*Toxoplasma gondii* antibodies (Table 1 and Table 2). The total sample size of the included studies was 6003 individuals (4356 pregnant women and 1647 HIV infected individuals). Of them, 4243 were found to be seropositive for *Toxoplasma gondii* infection. The overall apparent infection rate in pregnant women and HIV infected individuals was 66% and 84%, respectively.

**Table 1. List of included studies in meta-analysis on pregnant women.**

| Author | Study Year | Geographical Location | Regional State | Sampling method | Diagnostic test | Sample Size | Events | Event rate (AP) | LCI | UCI | QAS |
|--------|-----------|----------------------|----------------|-----------------|-----------------|-------------|--------|-----------------|-----|-----|-----|
| Biyansa [28] | 2019 | Northwest Ethiopia | Amhara | Systematic RS | LAT | 401 | 284 | 0.71 | 0.66 | 0.75 | 3 |
| Teweldemedhin et al. [29] | 2018 | Northern Ethiopia | Tigray | Simple RS | ELISA | 360 | 128 | 0.36 | 0.31 | 0.41 | 3 |
| Jula et al. [30] | 2015 | Southern Ethiopia | Oromia | Simple RS | EIA | 401 | 96 | 0.24 | 0.20 | 0.28 | 2 |
| Ahmed et al. [31] | 2012 | Central Ethiopia | Addis Ababa | Systematic RS | ELISA | 192 | 169 | 0.88 | 0.83 | 0.93 | 2 |
| Negero et al. [32] | 2016 | Southwest Ethiopia | SNNPR | Systematic RS | LAT | 210 | 159 | 0.76 | 0.7 | 0.82 | 3 |
| Negussie et al. [33] | 2014 | Southeast Ethiopia | Somali | Convenient | LAT | 301 | 201 | 0.67 | 0.61 | 0.72 | 2 |
| Yohanes et al. [34] | 2015 | Southern Ethiopia | SNNPR | Systematic RS | ELISA | 232 | 184 | 0.79 | 0.74 | 0.85 | 2 |
| Abamecha and Awel [35] | 2015 | Southwest Ethiopia | SNNPR | Systematic RS | ELISA | 232 | 198 | 0.85 | 0.81 | 0.9 | 3 |
| Agmas et al. [36] | 2013 | Northwest Ethiopia | Amhara | Systematic RS | LAT | 263 | 180 | 0.68 | 0.63 | 0.74 | 3 |
| Gelaye et al.[37] | 2010 | Central Ethiopia | Addis Ababa | Convenient | LAT | 288 | 246 | 0.85 | 0.81 | 0.89 | 3 |
| Awoke et al.[15] | 2013 | Northwest Ethiopia | Amhara | Simple RS | LAT | 384 | 71 | 0.18 | 0.15 | 0.22 | 2 |
| Hailu et al.[38] | 2012 | Selected parts of Ethiopia | Selected parts of Ethiopia | Simple RS | ELISA | 293 | 256 | 0.87 | 0.84 | 0.91 | 2 |
| Endris et al.[39] | 2011 | Northwest Ethiopia | Amhara | Convenient | LAT | 385 | 341 | 0.89 | 0.85 | 0.92 | 2 |
| Gebremedhin et al. [40] | 2011 | Central Ethiopia | Selected parts of Ethiopia | Simple RS | ELISA | 213 | 184 | 0.86 | 0.82 | 0.91 | 3 |
| Zemene et al.[41] | 2011 | Southwest Ethiopia | Oromia | Systematic RS | ELISA | 201 | 168 | 0.84 | 0.78 | 0.89 | 2 |

**Table 2. List of included studies in meta-analysis on HIV infected individuals.**

| Author | Study Year | Geographical Location | Regional State | Sampling method | Diagnostic test | Sample Size | Events | Event rate (AP) | LCI | UCI | QAS |
|---|---|---|---|---|---|---|---|---|---|---|---|
| Fewiza [42] | 2016 | Central Ethiopia | Addis Ababa | Convenient | ELISA | 174 | 99 | 0.57 | 0.5 | 0.64 | 3 |
| Zeleke and Melsew [43] | 2015 | Southwest Ethiopia | SNNPR | Systematic RS | ELISA | 270 | 255 | 0.94 | 0.92 | 0.97 | 2 |
| Tegegne et al.[44] | 2015 | Southwest Ethiopia | Oromia | Convenient | LAT | 135 | 109 | 0.81 | 0.74 | 0.87 | 3 |
| Yesuf and Melese [45] | 2014 | Southwest Ethiopia | Oromia | Systematic RS | - | 120 | 72 | 0.6 | 0.51 | 0.69 | 2 |
| Hailu et al.[38] | 2012 | Selected areas of Ethiopia | Selected areas of Ethiopia | Simple RS | ELISA | 190 | 178 | 0.94 | 0.9 | 0.97 | 2 |
| Yohanes et al.[46] | 2013 | Southern Ethiopia | SNNPR | Simple RS | ELISA | 170 | 150 | 0.88 | 0.83 | 0.93 | 2 |
| Walle et al.[47] | 2011 | Northwest Ethiopia | Amhara | Convenient | ELISA | 103 | 90 | 0.87 | 0.81 | 0.94 | 3 |
| Muluye et al.[48] | 2012 | Northern Ethiopia | Amhara | Systematic RS | LAT | 170 | 130 | 0.76 | 0.7 | 0.83 | 2 |
| Aleme et al.[49] | 2011 | Central Ethiopia | Addis Ababa | Simple RS | ELISA | 150 | 141 | 0.94 | 0.9 | 0.98 | 2 |
| Shimelis et al.[50] | 2007 | Central Ethiopia | Addis Ababa | Systematic RS | ELISA | 165 | 154 | 0.93 | 0.9 | 0.97 | 2 |

## Meta-analysis and bias assessment

The random effect model with inverse-variance procedures showed an overall pooled sero-prevalence of *Toxoplasma gondii* 72.5% (95% CI: 58.7% - 83.1%) in pregnant women and 85.7% (95% CI: 76.3% - 91.8%) in HIV infected people. A substantial heterogeneity among the studies on seropositivity for *Toxoplasma gondii* in pregnant women was evidenced ($I^2$ = 98.6%, Q-test = 970.24, df = 14, $p < 0.001$). Similarly, in HIV infected individuals, the heterogeneity amongst studies on anti-*Toxoplasma gondii* antibodies seropositivity was statistically significant ($I^2$ = 94.8%, Q-test = 172.68, df = 9, $p < 0.001$). The forest plots for each study group are shown in Figs 2 and 3. The Galbraith plot assessment amongst studies on anti-*Toxoplasma gondii* antibodies seropositivity in pregnant women and HIV infected people (Fig 4) also revealed that most of the included reports are laid outside of the 95% confidence limit and provided clear evidence of the variability of reports.

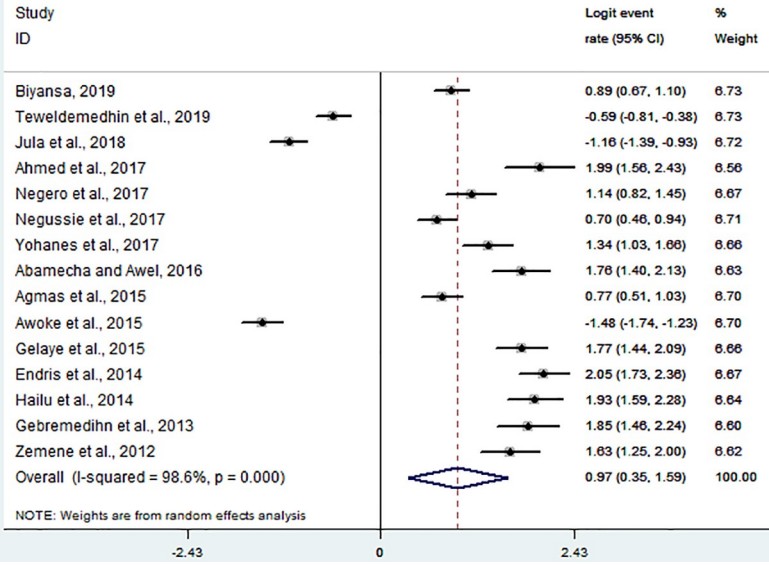

**Fig 2. Forest plot of the logit event rate estimate (lp) in pregnant women.**

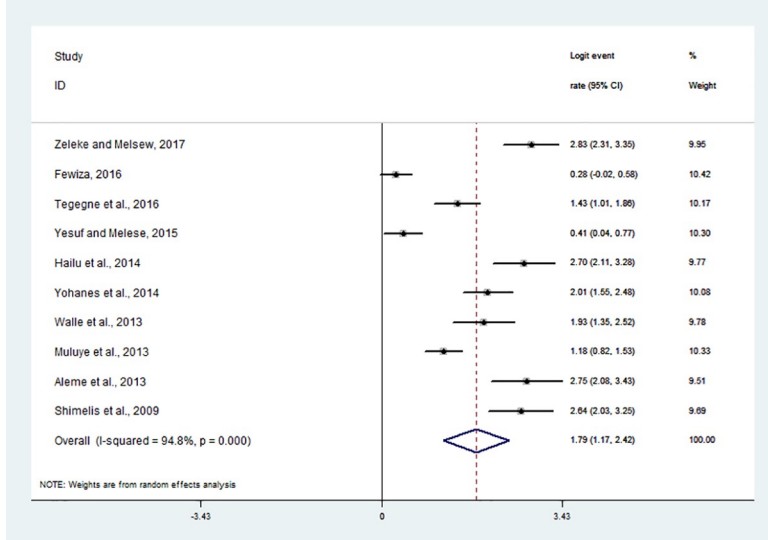

**Fig 3. Forest plot of the logit event rate estimate (lp) in HIV infected individuals.**

In our analysis, funnel plot observations (Figs 5, 6 and 7) and bias coefficients for studies published on anti- *Toxoplasma gondii* antibodies seropositivity in pregnant women (Egger's test: b = 26.24, 95% CI: 10.13% - 42.35%, *p* = 0.004 and Begg's test: *p* = 0.015) and HIV infected people (Egger's test: b = 14.8, 95% CI: 9.44% - 20.2%, *p* < 0.001 and Begg's test: *p* = 0.016) did confirm the presence of publication bias and small-study effects.

## Potential risk factors with seropositivity for *Toxoplasma gondii* in pregnant women

Nineteen potential risk factors (PRF) were identified and meta-analysis was made of thirteen of the fifteen included papers (Table 3). Of them, seven were shown a statistically significant effect on seropositivity for *Toxoplasma gondii* (test for the overall effect, *p* < 0.05) with higher odds of outcome: "history of abortion" (OR: 1.52), "age ≥ 35 years" (OR: 2.93), "contact with cat" (OR: 1.50), "cat ownership" (OR: 2.35), "knowledge about toxoplasmosis" (OR: 0.13), "being a housewife" (OR: 1.58) and "unsafe water source" (OR: 1.55) as shown in Table 3 and S1 Fig.

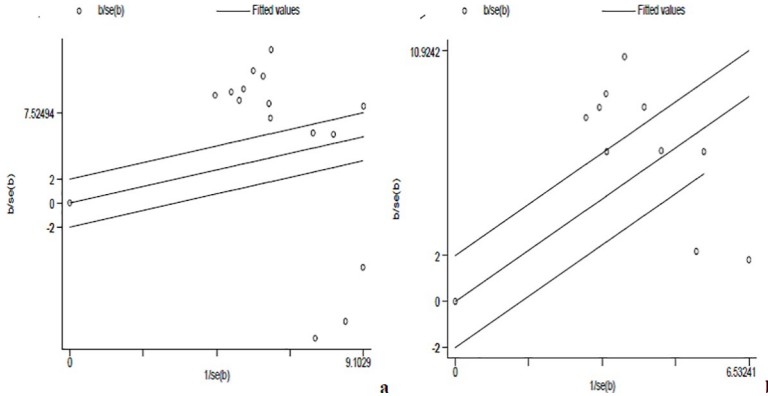

**Fig 4.** Galbraith plots for seroprevalence of toxoplasmosis in pregnant women (**a**) and HIV infected (**b**).

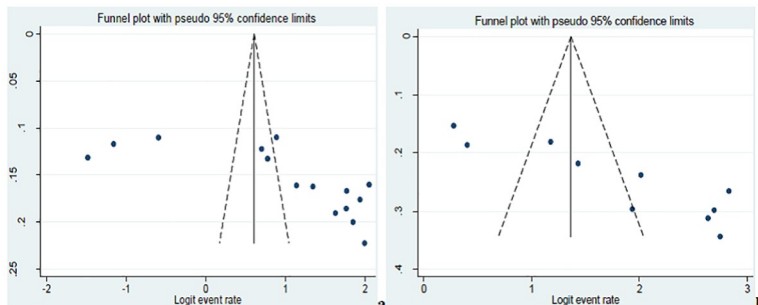

**Fig 5.** Funnel plots of the logit event rate of toxoplasmosis in pregnant women (a) and HIV infected (b).

An increased odds of seropositivity was also observed in eight assessed PRFs with *Toxoplasma gondii* infection among pregnant women; though these were not statistically significant ($p > 0.05$): "blood transfusion" (OR: 1.13), "contact with soil" (OR: 1.1), "dog ownership" (OR: 1.21), "being HIV positive" (OR: 1.22), "more than one pregnancy" (OR: 1.30), "raw meat consumption" (OR: 1.23), "raw vegetable consumption" (OR: 1.36) and "living in rural areas" (OR: 1.46). Further, we were not able to determine the effect of educational status, farming activity, raw milk consumption, and religion at odds of *Toxoplasma gondii* infection among pregnant women (Table 3). Forest plots, funnel plots and single weight of each publication contributing to the overall risk factors are presented in S1 Fig.

## Potential risk factors with seropositivity for *Toxoplasma gondii* in HIV infected individuals

Thirteen potential risk factors were identified and meta-analysis was performed on seven of the included ten papers (Table 4). Of them, three were shown a statistically significant combined effect on *Toxoplasma gondii* infection seropositivity (test for the overall effect, $p < 0.05$): "age $\geq 25$ years" (OR: 3.087), "cat ownership" (OR: 4.34) and "raw meat consumption" (OR: 2.43). On the other hand, six risk factors appeared to have higher odds of seropositivity but results have shown statistically insignificant overall effect ($p > 0.05$): "contact with cat" (OR: 1.37), "living as single" (OR: 1.20), "being illiterate" (OR: 1.1), "raw vegetable consumption" (OR: 1.14), "living in rural" (OR: 1.502) and "being female" (OR: 1.16) (Table 2). Besides, four risk factors: having "blood transfusion experience", having "knowledge about toxoplasmosis", "religion", and having "unsafe drinking water source" were not appearing to influence the odds of seropositivity among HIV infected individuals (Table 4). Forest plots, funnel plots and

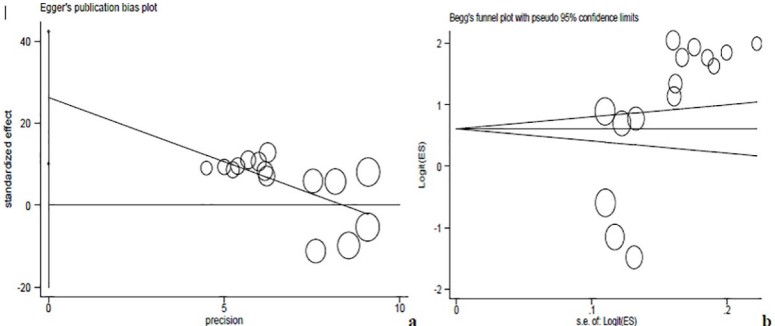

**Fig 6.** Eggers's publication bias plot (**a**) and Begg's funnel plot (**b**) reports for pregnant women studies.

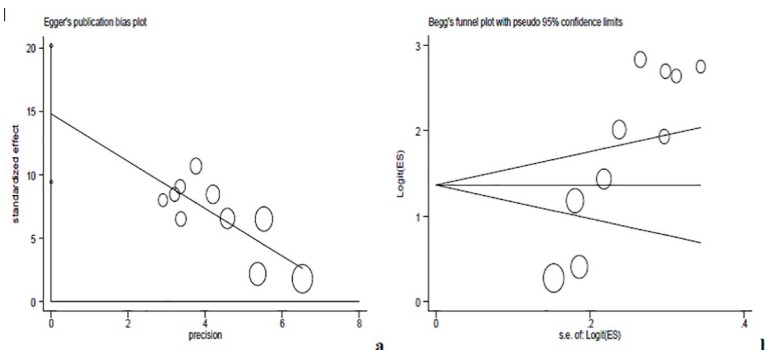

**Fig 7.** Eggers's publication bias plot (**a**) and Begg's funnel plot (**b**) reports for HIV infected individual studies.

single weight of each publication contributing to the overall risk factors are presented in S2 Fig.

## Discussion

### The pooled prevalence of seropositivity for *Toxoplasma gondii* infection

Ethiopia has diversified climatic conditions favoring the survival of different parasites (like *Toxoplasma gondii*) affecting domestic animals and humans. On the other hand, Ethiopian communities are categorized by low resources, including poor drinking water supply, coupled with different domestic animals (cats, dogs, goats, sheep, cattle, camels, and equines). Further, the communities are accustomed to raw or undercooked meat feeding habits and poor hygiene practices potentially increase the risk of *Toxoplasma gondii* infection. Considering the above situations, synthesizing information using previous studies is crucial, particularly among

**Table 3. Potential risk factors and their overall effects on *Toxoplasma gondii* seroprevalence in pregnant women in Ethiopia.**

| Potential risk factors | Effect model | OR(95%CI) | Overall effect | Num. studies | References ($p < 0.05$) | References ($p > 0.05$) |
|---|---|---|---|---|---|---|
| **History of abortion** | **IV, Fixed** | **1.52(1.15,2.00)** | **$p < 0.0001$** | **6** | [15,32] | [28,29,33, 37] |
| **Age ($\geq$35years)** | **IV, Random** | **2.93(1.73,4.95)** | **$p < 0.0001$** | **8** | **[29,32,36]** | **[15,30,31,34,37]** |
| Blood transfusion | IV, Fixed | 1.13(0.73,1.73) | $p = 0.59$ | 6 | | [15,28,29,32,34,37] |
| **Contact with cat** | **IV, Fixed** | **1.50(1.04,2.14)** | **$p = 0.03$** | **3** | **[35,36]** | **[30]** |
| **Cat ownership** | **IV, Random** | **2.35(1.44,3.85)** | **$p < 0.0001$** | **12** | **[15,29,31,32,36,39,41]** | **[28,33, 34,35,37]** |
| Contact with soil | IV, Fixed | 1.1(0.73,1.58) | $p = 0.72$ | 4 | [35] | [32,36, 41] |
| Dog presence | IV, Fixed | 1.21(0.86,1.71) | $p = 0.28$ | 3 | | [28,34,39] |
| Being illiterate | IV, Fixed | 1.02(0.83,1.24) | $p = 0.87$ | 12 | [29,35,36] | [15,28,30,32–34,37,39,41] |
| Farming/gardening activity | IV, Fixed | 0.70(0.46,1.08) | $p = 0.10$ | 2 | | [33, 34] |
| HIV status | IV, Fixed | 1.22(0.70,2.13) | $p = 0.47$ | 5 | | [31–33,37,39] |
| **Knowledge on toxoplasmosis** | **IV, Fixed** | **0.13(0.02,0.66)** | **$p = 0.01$** | **2** | | **[30,31]** |
| Number of pregnancy | IV, Random | 1.30(0.90,1.88) | $p = 0.17$ | 7 | [31,32] | [15,28,32, 34, 35] |
| **Being housewife** | **IV, Random** | **1.58(1.12,2.22)** | **$p = 0.01$** | **10** | **[29,36]** | **[28,30–35,37]** |
| Raw meat consumption | IV, Random | 1.23(0.77,1.98) | $p = 0.38$ | 12 | [15,31,32,34,35] | [28–30,33,37,39,41] |
| Raw milk consumption | IV, Fixed | 0.76(0.57,1.01) | $p = 0.06$ | 4 | | [28–30,32] |
| Raw vegetable consumption | IV, Random | 1.36(0.88,2.12) | $p = 0.17$ | 9 | [29,31,33,34] | [15, 28,30,33,37] |
| Christianity vs Muslim | IV, Fixed | 1.03(0.61,1.73) | $p = 0.91$ | 3 | | [31,33,39] |
| Residence: rural | IV, Random | 1.46(0.96,2.23) | $p = 0.08$ | 10 | [29,32,36] | [15,28,30,33–35,39] |
| **Unsafe water source** | **IV, Fixed** | **1.55(1.21,1.99)** | **$p < 0.0001$** | **9** | **[29,30,32]** | **[15,28,34,36,39,41]** |

**Table 4. Potential risk factors and their overall effects on *Toxoplasma* seroprevalence in HIV infected individuals.**

| Potential risk factor | Effect model | OR(95%CI) | Overall effect | Num. studies | References reported ($p < 0.05$) | References reported ($p > 0.05$) |
|---|---|---|---|---|---|---|
| **Age ($\geq$25yr)** | **IV, Fixed** | **3.087(1.36,6.99)** | ***p* = 0.0068** | **2** | **[46]** | **[44]** |
| Blood transfusion | IV, Fixed | 0.98 (0.42,2.32) | *p* = 0.97 | 4 | | [42,43,46,47] |
| **Cat ownership** | **IV, Fixed** | **4.34(2.49,7.56)** | ***p* < 0.0001** | **4** | **[42,44]** | **[46,49]** |
| Contact with cat | IV, Fixed | 1.37(0.74,2.54) | *p* = 0.31 | 3 | [47] | [43,48] |
| Being illiterate | IV, Fixed | 1.1(0.68,1.65) | *p* = 0.81 | 6 | | [43, 44, 46–49] |
| Knowledge on toxoplasmosis | IV, Fixed | 0.89(0.35,2.28) | *p* = 0.81 | 2 | [44,49] | |
| Marital status | IV, Fixed | 1.20(0.70,2.08) | *p* = 0.51 | 3 | | [46,48,49] |
| **Raw meat consumption** | **IV, Random** | **2.43(1.23,4.79)** | ***p* = 0.01** | **7** | **[44,46,47]** | **[42,43,48,49]** |
| Raw vegetable consumption | IV, Fixed | 1.14(0.73,1.78) | *p* = 0.56 | 6 | [49] | [42–44,46,47] |
| Religion (Christian) | IV, Fixed | 0.71(0.35,1.43) | *p* = 0.34 | 3 | | [44,48,49] |
| Residence (rural) | IV, Fixed | 1.50(0.83,2.73) | *p* = 0.18 | 5 | | [42,44,46–38] |
| Sex (female) | IV, Fixed | 1.16(0.58,2.32) | *p* = 0.67 | 6 | [44] | [42,46–49] |
| Unsafe water source | IV, Fixed | 0.80(0.37,1.74) | *p* = 0.57 | 3 | | [43,46,48] |

pregnant women and HIV infected individuals to depict the pooled prevalence and associated risk factors of *Toxoplasma gondii* infection. Estimating country-level and regional pooled prevalence with associated risk factors of *Toxoplasma gondii* infection could play a central role in developing suitable strategies for *Toxoplasma gondii* infection diagnosis, prevention, treatment, and control in Ethiopia, especially in pregnant women and HIV infected individuals. As far as the authors' knowledge, this study is the first to determine the potential risk factors of *Toxoplasma gondii* infection in pregnant women and HIV infected individuals of Ethiopia. Importantly, *Toxoplasma gondii* is widespread in Ethiopia among the general population and immunodeficient patients, and the high seroprevalence indicates a significant risk of clinical toxoplasmosis. But, there is a lack of comprehensive systematic and documented data on the potential risk factors contributing to *Toxoplasma gondii* infection in pregnant women and HIV infected individuals in Ethiopia.

In our study, the weighted overall prevalence of anti-*Toxoplasma gondii* antibodies was estimated to be higher in pregnant women (72.5%) and HIV infected individuals (85.7%). This suggests that the probability of high risk for congenital and cerebral toxoplasmosis in Ethiopia. Similarly, in previous studies, a higher prevalence of anti-*Toxoplasma gondii* antibodies was reported in Ethiopia: 81.4% of women of childbearing age [40], 80% of the factory workers [51] and 95.1% of hospitalized patients [52]. The pooled prevalence of anti-*Toxoplasma gondii* antibodies from studies administered to the general population in Ethiopia was also found to be 74% [53]. In contrast, the present finding is higher than the global pooled prevalence estimates in pregnant women (1.1% to 33.8%) [11,13] and HIV infected individuals (3.24% to 44.2%) [1,5,14]. Significantly lower pooled seroprevalence reports in pregnant women and HIV/AIDS patients were also reported in Mexico (15.62% & 20.2%) [54], Iran (41% & 50.1%) [55,56] and Nigeria (40.3% & 31.7%) [57]. This variation might be attributable to the difference in local climatic situations which determine the survival of oocysts and favours the dissemination and sporulation of oocysts, nutritional habit, the status of public health and sanitary services, personal hygiene, sources of drinking water, socio-cultural differences, residence, stray or pet cat density and management, and levels of close contact with cats [1,7,13,17,58–60]. Several studies have also shown the effects of close contact with the meat of infected animals, consumption of raw meat, vegetable, and contaminated water [1,13,56,61–64]. Further, the infection rate has been reported to be influenced by education level, socioeconomic status, age groups, and local religious beliefs of the community [1,7,58,62,65,66].

## Potential risk factor analysis on seropositivity for *Toxoplasma gondii*

The epidemiology of *Toxoplasma gondii* infection is dependent on various environmental factors (food habits, climate, sanitary status and contact with infected cat faeces), sociodemographic factors (age, sex, occupation, education, economic status, and residence) and parasite factors (virulence, genotype or behaviour). Of these factors, one has little effect on the epidemiological scheme of *Toxoplasma gondii* infection, but together, they can impact the distribution pattern of the disease in the globe and even within the same country [56,67].

Age is an important sociodemographic factor associated with *Toxoplasma gondii* infection. We observed significantly higher odds of *Toxoplasma gondii* infection in pregnant women and HIV infected individuals as the age of study subjects increasing. This in line with the studies that reported an increasing *Toxoplasma gondii* seropositivity with increasing age of study subjects [7,13,19,58,63,68]. This is also supported by various authors in the globe [67,69–74]. The studies included in this review were also considered age as a potential factor [29,32,36,46]. This might be explained by the assenting interaction between an increase in age, with a prolonged risk of exposure to *Toxoplasma gondii* oocysts and viable tissue cysts (bradyzoites) from the meat of infected animals over time, with the long-lived immune response, the availability of divers transmission route and lack of community awareness [13,67,75]. This could also be due to that the older group could have a longer period of exposure to any of the risk factors [13,76,77]. Further, this might be due to the increasing tendency of the people to eat raw or undercooked meats, fast foods (burgers and sausages), and/or increasing close contact with pet cats as their age increases.

*Toxoplasma gondii* seropositivity has shown an association with the history of abortion ($p < 0.05$). This is consistent with the previous reports [76,78–80]. Cats play a central role in the epidemiology of *Toxoplasma gondii* and are major sources of viable oocysts for environmental contamination [3,7,74]. They can acquire the infection through the consumption of raw meat with tissue cysts (bradyzoites) from infected animals. After ingestion of one viable tissue cyst, cats can release millions of viable oocysts and increase the likelihood of pathogen transmission to the risk groups [56,63]. Being cat ownership, close contact with cats and abundance of cats are also considered to be the important drivers for *Toxoplasma gondii* infection in humans [17,58,74]. In this study, ownership of cats was found with a significantly higher combined odds ratio in both study targets. Contact with cats was also found with a significantly higher odds ratio in pregnant women. This suggests that ownership of cats or close contact with cats, coupled with frequent exposure to cat faeces, cats' litter box management way or neglect of preventive measures (hand washing or wearing gloves) could increase the risk of infection to an appreciable level [17,40,56,62,67,74].

*Toxoplasma gondii* infection in high-risk groups such as pregnant women, HIV infected individuals, and cancer patients has been reported to be influenced by educational status and knowledge about toxoplasmosis [58,74]. Liu et al. [62] and Sun et al. [81] also stated that individuals with no knowledge and formal education are more likely to acquire *Toxoplasma gondii* infection. Other authors suggest that health education as a cost-effective intervention strategy for *Toxoplasma gondii* infection [17,62,74]. This is supported by our findings in which literacy and health knowledge are considered as protective factors. Similar findings also reported by other authors [43,44,62,63]. This might be attributed to that those with low/no formal education status may have less hygienic practice and they are more likely to acquire *Toxoplasma gondii* infection [58,62]. Lack of basic information about toxoplasmosis such as a source of infection, hygiene, raw meat/vegetable consumption, transmission route, and ignorance of the disease can maintain the risk of infection.

Previous reports have pointed out that *Toxoplasma gondii* seroprevalence varies between various working groups. Certain working groups have more contact than others with

*Toxoplasma gondii* directly or indirectly. The present meta-analysis reveals that housewives are a more vulnerable group with significantly higher *Toxoplasma gondii* seroprevalence. Our results are in line with those studies done in Saudi Arabia and the United States [61,82,83]. Habitually, housewives spend more time taking care of pets, cooking, and tasting food at home during meal preparation, handling and chopping meat without wearing gloves, cleaning and washing vegetables, and engaging in gardening, mainly in the rural regions [64,67].

Published reports have documented that unhygienic drinking water is a considerable risk factor for *Toxoplasma gondii* infection in humans and animals [17,74]. In our review, unsafe drinking water has also shown a significant association with *Toxoplasma gondii* infection in pregnant women but not in HIV infected individuals. This might be due to the contradicting findings of the included studies and wide confidence intervals. Hence, we considered the absence of tap water as an indicator of unsafe drinking water quality. However, this might not be necessarily true. The microbiological quality of well or spring water could be satisfactory for the majority of the time unless contamination events happen by flooding and contaminated animals [84,85].

Higher odds of *Toxoplasma gondii* infection was confirmed in pregnant women with risk factors like raw meat and vegetable consumption; though it was not significant. Similarly, raw meat consumption habit has shown significantly higher odds of *Toxoplasma gondii* infection in HIV infected individuals (*p* = 0.01). This is consistent with reports across the globe [37,62,74,86–88]. The consumption of raw meat with tissue cysts and vegetables contaminated with sporulated oocysts are considerable sources of infection [63]. The risks of acquiring *Toxoplasma gondii* via contaminated meat with tissue cysts and vegetables with oocysts from contaminated soil and water vary with culture and feeding habits in various communities [17].

This systematic review and meta-analysis has certain limitations, including: (1) all studies were used cross-sectional study design, this may bias our risk factor analysis and hence, the result should be interpreted with caution; (2) all studies that exploring seroprevalence/ factors were drawn from limited samples of participants which may not represent the national seroprevalence, therefore extrapolative values and estimates for the identified risk factors should be assessed holistically; (3) lack of related risk factors evaluated by the majority of papers; (4) the number of eligible studies used for each risk factor analysis were also small; so the estimates of the risk factors were made accordingly; it might decrease the power of meta-analysis; (5) limited or lack of studies in many regions of Ethiopia, which reduce the representativeness of the pooled estimate; (6) lack of a standard questionnaire to gather suitable data in the country; (7) studies were performed with different serological tests without similar sensitivities and specificities, and (8) the results of available data were heterogeneous. However, the finding provides an insight into the prevalence of *Toxoplasma gondii* infection and associated risk factors and may serve as reference paper to plan and establish control measures among pregnant women and other immunocompromised patients in Ethiopia.

In conclusion, as far as to the knowledge of the authors, this is the first systematic review and meta-analysis of the risk factors associated with seropositivity for *Toxoplasma gondii* among pregnant women and HIV infected individuals in Ethiopia. The results of this study showed that pregnant women and HIV infected individuals have a high risk of exposure to *Toxoplasma gondii* infection. The pooled prevalences of anti- *Toxoplasma gondii* antibodies were 72.5% in pregnant women and 85.7% in HIV infected individuals. This suggests that thousands of immunocompromised patients are at risk of toxoplasmosis due to the socio-cultural and living standards of the communities of Ethiopia. Risk factors (age, abortion history, contact with cats, cat ownership, having knowledge about toxoplasmosis, being a housewife and having unsafe water source) were shown significant overall effects on *Toxoplasma gondii* infection rate in pregnant women. Age, cat ownership, and raw meat consumption also

showed significant overall effects on infection rate with *Toxoplasma gondii* in HIV infected individuals. The results of this study are appreciated to increase awareness among public health workers and educators regarding *Toxoplasma gondii* infection in high-risk groups (pregnant women and HIV infected individuals). Appropriate preventive measures are needed to reduce the exposure to *Toxoplasma gondii* infection. Further studies to investigate important risk factors are recommended to support the development of more cost-effective preventive strategies.

## Ethical approval

None sought.

## Supporting information

**S1 Text. PRISMA-checklist.**
(DOC)

**S1 Data. Extracted data file.**
(XLS)

**S1 Table. STROBE Statement-checklist (risk of bias criteria).**
(DOC)

**S1 Fig. Potential risk factors with estimated odds ratio and funnel plots for *T. gondii* seropositivity in pregnant women of Ethiopia.**
(PDF)

**S2 Fig. Potential risk factors with estimated odds ratio and funnel plots for *T. gondii* seropositivity in HIV infected individuals of Ethiopia.**
(PDF)

## Author Contributions

**Conceptualization:** Zewdu Seyoum Tarekegn, Haileyesus Dejene.

**Data curation:** Agerie Addisu, Shimelis Dagnachew.

**Formal analysis:** Zewdu Seyoum Tarekegn, Haileyesus Dejene.

**Investigation:** Agerie Addisu, Shimelis Dagnachew.

**Methodology:** Zewdu Seyoum Tarekegn, Haileyesus Dejene.

**Software:** Zewdu Seyoum Tarekegn, Haileyesus Dejene.

**Validation:** Agerie Addisu, Shimelis Dagnachew.

**Visualization:** Zewdu Seyoum Tarekegn, Haileyesus Dejene, Agerie Addisu, Shimelis Dagnachew.

**Writing – original draft:** Zewdu Seyoum Tarekegn, Haileyesus Dejene.

**Writing – review & editing:** Agerie Addisu, Shimelis Dagnachew.

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
