## [Decision Letter · Decision Letter 0]

18 Jun 2020

Dear Dr. Tarekegn,

Thank you very much for submitting your manuscript "Potential risk factors associated with Toxoplasma gondii infection in pregnant women and HIV infected individuals in Ethiopia: A systematic review and Meta-analysis" for consideration at PLOS Neglected Tropical Diseases. As with all papers reviewed by the journal, your manuscript was reviewed by members of the editorial board and by several independent reviewers. In light of the reviews (below this email), we would like to invite the resubmission of a significantly-revised version that takes into account the comments. 

We cannot make any decision about publication until we have seen the revised manuscript and your response to the reviewers' comments. Your revised manuscript is also likely to be sent to reviewers for further evaluation.

Sincerely,

Olaf Horstick, FFPH(UK)

Associate Editor

Pikka Jokelainen

Deputy Editor

Reviewer #1:

This is a work of high importance and it is generally well conducted. I have a couple of minor but essential comments that should be addressed. They can be found as comments in the attached document.

Reviewer #3: The main problem (in my opinion) for this study is novelty of results, as we know there are some recent global and regional meta-analyses considering prevalence and risk factors of T. gondii in pregnant women and HIV patients; and data from Ethiopia is present in them.

Introduction

* There are four global studies with regard to acute and latent toxoplasmosis in pregnant women (https://doi.org/10.1371/journal.pntd.0007807 and https://doi.org/10.1016/j.cmi.2020.01.008) and in HIV patients (10.1097/QAD.0000000000002424 and http://dx.doi.org/10.1016/S2352-3018(17)30005-X). I think use of results of these studies is informative for introduction. Moreover in discussion results of present study should be compared with global and regional prevalences according to results from abovementioned studies. 

Methods

* Data extraction: please mention details for risk factors.

* Meta-analysis: the studies by Foroutan-Rad et al. and Khalkhali et al. could not be a protocol. Please refer to original study by DerSimonian and Laird (1986).

* In following sentence, used reference is invalid and authors should add valid reference. "Cochran's Q-statistics and inverse variance index (I2) were computed to determine the heterogeneity and inconsistency among studies, respectively [2]."

* I think only random effects model is proper for estimation of pooled prevalence rates. 

Results 

* 3.2. Meta-analysis and bias assessment: please add 95% CI related with pooled prevalences.

* Results for prevalence rates presented in text is inconsistent with forest plots (figure 2 and 3).

* All fig are numbered as figure 1. Please correct them.

There are several editorial errors that should be modified in revise.

Deputy Editor: 

The Reviewers find the work interesting but there are several points that need to be clarified before the work can be fully evaluated. Based on the evaluation of the work by three Reviewers, Associate Editor and myself, the decision is ‘major revision’. We are looking forward to receiving the revised manuscript. 

Comments from Deputy Editor:

The author summary should be a summary of this work. I recommend to largely rewrite the current summary. 

The conclusions and recommendations given must be based on this study. What is meant by ‘burden’ in the first concluding sentence? Can it be concluded directly from this study that routine serological screening and health promotion should be started? Can the aspects about cats mentioned in the last paragraph be concluded from this study? 

This study was limited to one country. Have similar studies been made in other countries? I expect in most countries, the number of studies conducted is too low for a similar study – this could be discussed. 

Relevant basic background information about the setting of the study (e.g. population of Ethiopia) should be given. It would be good to emphasize why this work was important to do, in particular the importance to international audience. It is important that the setting is repeated in the Discussion, to make it clear the results are from studies performed in one country. E.g. the first sentence of the last paragraph should specify the setting. 

Limitations of the study need to be discussed more. 

Please check the use of the key expressions everywhere in the manuscript very carefully. In particular, ‘toxoplasmosis’ is usually used for the clinical disease; for the subclinical infection detected by serology, ‘T. gondii infection’ is better expression. For example, please change ‘Seroprevalence of toxoplasmosis’-> T. gondii seroprevalence; and please rephrase ‘95% confidence intervals (Cis) of toxoplasmosis’. The name of the parasite should be written in full (Toxoplasma gondii, in Italics)

Please check that abbreviations are explained when used for the first time (PRF). 

Avoid using the word ‘only’ in results.

Table 1: Sampling method ‘simple’ should be explained. 

The variable ‘age’ was dichotomized; this needs to be mentioned in Methods.

Consider changing ‘Ethiopian’ to ‘in Ethiopia’, if the inclusion criteria of the original studies did not include the nationality of the individuals.

In the Discussion, the sentence starting ‘Higher Toxoplasma seroprevalence was reported’ is confusing, as two of the three estimates mentioned are not higher than the pooled estimates obtained. Ref nr 48 was not included in the study, but it is used for comparison. Please check/clarify. 

In the Discussion, the sentence starting ‘Comparable findings…’ puts the study in the global setting, but, the references used (nr 52-58) do not cover different areas across the world very well. In some parts of the world, the seroprevalence is clearly lower. I can see an opportunity to widen the interest of this work by including a wider scope here. 

 ‘infected meat’ -> meat of infected animals

‘also depends on’ -> has also been associated with

‘infection… has been reported to be influenced by’ – please edit, the infection was likely not shown to be influenced. 

The sentences providing possible explanations for the higher seroprevalence in older age groups needs checking and editing. Is it reasoned to mention only oocysts? 

The discussion about cats would benefit from editing; please check that the sentences are supported by the references. 

Please rephrase the sentence using expression ‘contaminated individuals/animals’.

Please make sure the figure legends provide enough information so that the figures can also “stand alone” without the text. E.g. the parasite and the country should be mentioned

Was the quality of the included studies evaluated in any way? If not, why? 

Including the raw data that was extracted as supplementary material would be a very good addition. 
---

## [Decision Letter · Decision Letter 1]

15 Sep 2020

Dear Dr. Tarekegn,

Thank you very much for submitting your manuscript "Potential risk factors associated with seropositivity for Toxoplasma gondii among pregnant women and HIV infected individuals in Ethiopia: A systematic review and Meta-analysis" for consideration at PLOS Neglected Tropical Diseases. As with all papers reviewed by the journal, your manuscript was reviewed by members of the editorial board and by several independent reviewers. The reviewers appreciated the attention to an important topic. Based on the reviews, we are likely to accept this manuscript for publication, providing that you modify the manuscript according to the review recommendations. 

Sincerely,

Olaf Horstick, FFPH(UK)

Associate Editor

Pikka Jokelainen

Deputy Editor

Reviewer's Responses to Questions

**Key Review Criteria Required for Acceptance?**

**Methods**

-Are the objectives of the study clearly articulated with a clear testable hypothesis stated?

-Is the study design appropriate to address the stated objectives?

-Is the population clearly described and appropriate for the hypothesis being tested?

-Is the sample size sufficient to ensure adequate power to address the hypothesis being tested?

-Were correct statistical analysis used to support conclusions?

-Are there concerns about ethical or regulatory requirements being met?

Reviewer #1: (No Response)

Reviewer #3: (No Response)

**Results**

-Does the analysis presented match the analysis plan?

-Are the results clearly and completely presented?

-Are the figures (Tables, Images) of sufficient quality for clarity?

Reviewer #1: (No Response)

Reviewer #3: (No Response)

**Conclusions**

-Are the conclusions supported by the data presented?

-Are the limitations of analysis clearly described?

-Do the authors discuss how these data can be helpful to advance our understanding of the topic under study?

-Is public health relevance addressed?

Reviewer #1: (No Response)

Reviewer #3: (No Response)

**Editorial and Data Presentation Modifications?**

Reviewer #1: (No Response)

Reviewer #3: (No Response)

**Summary and General Comments**

Reviewer #1: This looks much better now compared to before the corrections made. However, I miss a sound justification why this systematic review was conducted in addition to those already present, which were mentioned by one of the reviewers. Within the introduction section it still reads: "no systematic studies have been conducted to determine the prevalence with associated risk factors of Toxoplasma gondii infection in pregnant women and HIV infected individuals." 

Once this issue is convincingly addressed, I would recommend this SLR for publication.

Reviewer #3: (No Response)

PLOS authors have the option to publish the peer review history of their article (what does this mean?). If published, this will include your full peer review and any attached files.

Reviewer #1: No

Reviewer #3: No
---

## [Editor Report · Decision Letter 2]

3 Nov 2020

Dear Dr. Tarekegn,

We are pleased to inform you that your manuscript 'Potential risk factors associated with seropositivity for Toxoplasma gondii among pregnant women and HIV infected individuals in Ethiopia: A systematic review and Meta-analysis' has been provisionally accepted for publication in PLOS Neglected Tropical Diseases.

Best regards,

Olaf Horstick, FFPH(UK)

Associate Editor

Pikka Jokelainen

Deputy Editor

---

## [Editor Report · Acceptance letter]

2 Dec 2020

Dear Dr. Tarekegn,

We are delighted to inform you that your manuscript, "Potential risk factors associated with seropositivity for Toxoplasma gondii among pregnant women and HIV infected individuals in Ethiopia: A systematic review and Meta-analysis," has been formally accepted for publication in PLOS Neglected Tropical Diseases.

Best regards,

Shaden Kamhawi

co-Editor-in-Chief

Paul Brindley

co-Editor-in-Chief
